# Smoking Among Healthcare Professionals in Australia: A Scoping Review

**DOI:** 10.3390/ijerph22010113

**Published:** 2025-01-15

**Authors:** Masudus Salehin, Louisa Lam, Muhammad Aziz Rahman

**Affiliations:** 1Institute of Health and Wellbeing, Federation University Australia, Berwick, VIC 3806, Australia; louisa.lam@acu.edu.au; 2Collaborative Evaluation and Research Centre (CERC), Federation University Australia, Berwick, VIC 3806, Australia; 3School of Nursing, Midwifery and Paramedicine (VIC), Faculty of Health Sciences, Australian Catholic University, Fitzroy, VIC 3065, Australia; 4School of Public Health and Preventive Medicine, Monash University, Melbourne, VIC 3004, Australia

**Keywords:** smoking, prevalence, predictors, health professionals, Australia

## Abstract

Studies showed healthcare professionals who are non-smokers are more likely to deliver smoking cessation advice to their patients than those who are smokers. However, healthcare professionals continue to smoke across the globe. This scoping review assessed the available data on the prevalence and predictors of smoking among healthcare professionals in Australia. Following the PRISMA extension for the Scoping Review checklist, a systematic literature search was conducted on CINAHL, MEDLINE, APA PsycINFO, Scopus, Web of Science, and Cochrane Library in August 2024. Articles published between 1990 and 2024 were considered, and finally, 26 papers met the inclusion and exclusion criteria. Australian healthcare professionals showed varying smoking prevalence. For physicians, it was 10.2% in 1990 to 7.4% in 2013; among dentists, 6% in 1993 to 4.9% in 2004; and among nurses, 21.7% in 1991 and 10.3% during 2014–15. The highest smoking rates were observed among Aboriginal health workers (AHWs): 63.6% in 1995 to 24.6% in 2021. Age was a positive predictor for smoking among nurses, and so was male gender among dentists, physicians, and nurses; other predictors included area of specialty, lower emotional wellbeing, etc. This review highlighted a declining trend in smoking among healthcare professionals in Australia; however, it was not proportionate among the different health specialties.

## 1. Introduction

The Global Burden of Disease study states smoking is the third leading risk factor for deaths and disability in the world [1]. However, of all people, on a global level, health professionals are not immune from smoking. Like others, healthcare workers also have a tendency to smoke [2] despite being aware of its well-known deleterious effect on health, and also on their image as role models to patients [3]. The influence of smoking habits among healthcare professionals can compromise their public perception and health promotion roles, as their professional position may at times conflict with their personal choice of smoking [4]. Studies have revealed statistically significant associations between physicians’ smoking status and beliefs and their clinical practice [5,6].

There is also evidence that smoking cessation advice from a health professional has a positive effect on their patients or community [7,8], as smokers who rely on the support and advice of their healthcare provider have more chances to quit than those who try it on their own [6,9]. The WHO Framework Convention on Tobacco Control (FCTC) also specifically stresses the importance of healthcare workers setting an example by not using tobacco [10]. Smoking by health professionals may then undermine their health promotion role and send a key message to smokers that smoking is a healthy choice [11].

In the same context, it is also important to understand why healthcare workers continue to smoke and what the drivers for such behavior are. It was not unfamiliar even in the earlier days to use physicians/health workers in cigarette advertisements. Through the early 1950s, it was a strategic response by the tobacco companies to devise advertising directly to physicians; the doctors’ image was good for them (tobacco industries) to normalize smoking and also to assure the consumer that their brands were safe [12]. Indeed, the majority of the physicians used to smoke then, as described in the article ‘The Doctors’ Choice is America’s Choice!’ [12]. But why would physicians, nurses, or health workers per se continue to smoke despite knowing smoking is harmful? Is it just an addiction or enjoyment that drives them like the general people [13], or perhaps they consider it important, but not a priority [14]? Studies would narrate, among other reasons, a possible stressful working environment, peer pressure, socioeconomic status, or education as some of the possible predictors [14,15]. It is imperative to know whether these and other variables in regard to smoking change over time, place, or geographic locations because they might provide a behavioral/motivational ladder to locate their blockage for change and/or design help specifically addressed to them [14].

Historically, there has been a decline in smoking rates among healthcare professionals in many countries. However, a recent systematic review and meta-analysis on the prevalence of tobacco use among healthcare workers showed varying degrees of smoking prevalence across nations. It showed that among high-income countries, the mean smoking prevalence among healthcare workers was lower than the general population, except in Australia, Italy, and Uruguay [10]. The lowest prevalences (<5%) were observed in the US and Ireland, and the highest were (30%) in Greece, Croatia, Italy, and Uruguay. Among upper-middle-income countries, the lowest prevalences (<30%) among healthcare workers were observed in Argentina, Brazil, and Mexico, and the highest was (40%) in Turkey [10]. Among lower-middle-income countries, the lowest smoking prevalence (<10%) was noted in India and the highest (>50%) in Pakistan [10]. So, despite all the anti-smoking campaigns and policies and local, national, and international legislations and commitments, the decline in tobacco smoking prevalence has not been symmetrical or proportional across regions, including those among the healthcare groups. Public health efforts have also historically focused on smoking cessation programs for the general population . Healthcare workers, who are at the forefront of providing smoking cessation counseling services to the patients or community, have not been a focus of tobacco control initiatives [4,16]. Also, when we talk about interactions with patients on smoking cessation, it is not only the physicians or nurses but an array of other health service provider groups as well. However, evidence on smoking prevalence or their correlates as such for this wider group of healthcare workers is generally lacking. The existing literature is either based on sporadic cross-sectional studies or, in some cases, analyses from longitudinal data on smoking prevalence. Australia, in this regard, is not an exception either.

Australia has a unique healthcare environment. Although physicians and nurses constitute the two largest professional groups in the Australian healthcare system [11,17], there are other health service providers who are well positioned to offer smoking cessation advice to patients. They include dentists, midwives, psychologists, optometrists, occupational therapists, mental health workers, pharmacists, physician assistants, and Aboriginal and Torres Strait Islander health practitioners (AHWs), as they are on the frontlines of primary care [18,19]. The importance of the AHWs should not be undermined, as they play a crucial role in delivering a range of services and health information to the Indigenous communities [20]. They are a subgroup often overlooked in research but who nevertheless face daunting challenges in regard to smoking cessation or the cultural context of smoking within Aboriginal communities.

Although Australia is one of the countries where smoking rates have declined consistently and considerably over the years because of its very early and stringent tobacco control policies and legislations, like many other countries, information from across the employee spectrum of the Australian health workforce is very limited [21]. Nationally representative surveys in Australia, such as the National Health Surveys (NHS) or the National Drug Strategy and Household Surveys (NDSHS), do not provide statistical information for disaggregated groups of people that represent healthcare providers [22]. Combating tobacco smoking among health professionals requires the availability of current data on their smoking behavior. Factoring smoking prevalence trends and possible reasons for such behavior and their impact on health efficacy, recent public health campaigns, and technological advancements in data collection and shifting societal attitudes toward smoking in healthcare, it is imperative that we assess the smoking behavior among the wider health professional groups in Australia.

To address this gap in research, we aimed to undertake a comprehensive scoping review. The primary objective was to explore the prevalence of smoking among healthcare professionals in Australia. The secondary objective was to identify the factors influencing the smoking habits among these health professionals. The uniqueness of this review will be the novel methodological perspective, including the wide range of workforce in the country and underrepresented groups like AHWs. Addressing the gaps may also lead to some actionable outcomes, such as informed tailored interventions or policy changes. To recognize an evidence-based literature range, and also follow the PCC (population, concept, and context) framework [23], this scoping review’s research question was developed as follows:

What does the academic literature say about the prevalence and predictors of smoking among healthcare professionals in Australia?

## 2. Materials and Methods

### 2.1. Review Design

A systematic review is considered to be the pillar of evidence-based healthcare [24]. However, based on our study aim and considering the limited resources required for the rigorous systematic review process, a scoping review was determined to be the most appropriate method. A scoping review is a very robust tool to explore the scope/coverage of a body of literature on a given topic, as it allows one to map the literature [24]. To ensure best practice methods, we used the PRISMA-ScR (Preferred Reporting Items for Systematic Reviews and Meta-Analyses (PRISMA) extension for scoping reviews) checklist [25,26,27]; the PRISMA checklist was not registered though. To ensure robustness, this review was also guided by the methodological framework for scoping reviews of Arksey and O’Malley and is consistent with Levac et al.’s scoping review guidelines [27]. Levac et al. identified six steps in conducting a scoping review: (i) identify the research question; (ii) identify relevant literature; (iii) select studies; (iv) chart studies; and (v) collate, summarize, and report the studies [28].

### 2.2. Eligibility Criteria

The inclusion and exclusion criteria were based on the Population, Concept, and Context framework (Table 1). The literature search was limited to English-language articles. Articles published between 1990 and 2024 were examined. This timeframe coincided with the timing of the National Health Survey (NHS) conducted by the Australian Bureau of Statistics (ABS), which was initiated in 1989 to acquire information on the national health status of Australians [29]. The 1990s also marked the time when Australia banned smoking in workplaces and public places and imposed a ban on advertising tobacco products in print media; this ushered in a decline in tobacco consumption in the country [30] . Studies were disregarded if they did not meet the inclusion criteria (Table 1). We also reviewed reference lists of all qualified articles to identify studies that may have been missed during the database searches. As part of the inclusion criteria, a few terms were applied to carry out the literature search. *Smoking of tobacco* was referred to as the consumption of cigarettes, cigars, bidis, electronic cigarettes (also referred to as vape), pipes, water pipes (also referred to as hookah, shisha, narghile, or argileh). *Smoking prevalence* estimation was based on the number of ’current smokers’ among the study population [31]. Data were grouped into two tables for synthesis: one for smoking prevalence and the other for predictors of smoking among healthcare professionals in Australia.

### 2.3. Databases

A systematic literature search of six electronic databases, CINAHL Complete, MEDLINE, APA PsycINFO, Scopus, Web of Science, and Cochrane Library, was performed from 1 to 15 August 2024. These databases were selected as they comprehensively capture the nursing, medical, and psychological literature on smoking globally and are often used extensively. MEDLINE and CINAHL were searched using OVID and EBSCO host interfaces, respectively, and Web of Science, Scopus, and Cochrane Library were searched using their own web interface. Additionally, an in-depth exploration of reference lists of related papers and gray literature was hand-searched to identify studies and reports of relevance to this review.

### 2.4. Search Strategy for Electronic Databases

The search terms were identified by the research team in consultation with a research librarian and used best practice techniques for searching databases (e.g., capturing synonyms with MeSH terms in MEDLINE). Efforts were made to design the database search strategy to be as comprehensive as possible. Based on the PCC framework [23], search terms for four constructs of interest were identified first, namely ‘smoking’, ‘prevalence’, ‘predictors of smoking’, ‘health professionals’, and ‘Australian’. All identified keywords and index terms were used to formulate search strategies specific to the selected databases. Appropriate truncation symbols were used to account for search term variations and maximize searches. Initially, each term within a construct was searched individually, and then these results were combined with a Boolean ‘AND’ criterion (Appendix A).

### 2.5. Data Selection and Extraction

Titles and abstracts of records were downloaded and imported into EndNote bibliographic software and from there to the Covidence online tool to streamline our systematic scoping review process. All duplicates were automatically removed once uploaded to Covidence by the de-duplication process and by hand. At first, titles and abstracts were screened against the inclusion criteria, and then potentially relevant papers went through a full-text review. The research question, being a broad one, allowed the search to identify many irrelevant articles, systematic reviews, and gray literature, which although excluded from our scoping review, nevertheless contributed to comprehensive understanding and background knowledge. The first author (MS) screened all titles and abstracts, and half of the titles and abstracts were double-screened by the other two researchers (MAR and LL). In case both the reviewers decided to exclude a study, they were required to specify a similar reason for exclusion. This ensured no relevant paper was being screened out inadvertently. As a consistency check, full-text-level reviewing was independently carried out by MS and LL with conflicts/disagreements resolved through consensus by the third reviewer, MAR. Relevant data were extracted using a standardized data extraction and quality assessment criteria form. Percentage agreement was set to 80% to ensure inter-rater reliability; this was considered a satisfactory level of adherence to selection criteria. Charting of data was manually performed by extracting the data in a table from the finally selected full-text articles. Two main tables were constructed for the two variables: one for the prevalence of smoking (categories included author, publication year, geographical location, study population, research design, main results, study limitations, etc.) and another for predictors of smoking. The latter one was further clustered under three variables, namely biological and demographic, psychological/psychosocial, and environmental predictors. Coding keys were created in accordance with the Population, Concept, and Context framework; prevalence and predictors were tabulated for each variable, according to the coding keys. Statistical analysis was not conducted due to a lack of required statistics; these included incomplete or absent information in regard to *p*-values, odds ratios, or 95% confidence interval levels to examine statistical associations.

### 2.6. Risk of Bias/Quality of the Selected Studies

The methodological quality of the selected studies for this review was assessed using the adapted Newcastle Ottawa Scale (Appendix B), which is appropriate for non-randomized studies.

## 3. Results

### 3.1. Study Selection

Our initial search identified 2930 references. Five additional records were added from reference list checks and hand searches. After the removal of duplicates (1485), we were left with 1450 potentially relevant papers (Figure 1). Following the review of titles and abstracts, and applying the selection criteria, we were able to retrieve 88 full papers for a more detailed screening. From these, twenty-six papers were finally included that fulfilled the eligibility criteria. While all of the selected studies provided evidence on smoking prevalence, only twelve studies explored predictors of their smoking behavior (Table 2 and Table 3).

### 3.2. Study Characteristics

The majority of the studies (*n* = 21) in this review were cross-sectional in design [20,21,32,33,34,35,37,38,40,41,42,43,44,46,47,49,50,51,52,53,54], and two were secondary data analyses from cross-sectional studies conducted earlier [11,36]. Of the remaining studies, two adopted mixed-method study designs [39,48], and the other one was descriptive exploratory [45] (Table 2). Most of the included studies (*n* = 16) adhered to postal/telephone surveys or face-to-face interviews for data collection [11,20,21,32,33,34,35,38,39,44,46,47,48,49,50,54]: four used online surveys [37,41,42,53], two used a combination of both postal and online surveys [36,51], and the remaining three used focus group discussions (in conjunction with postal and/or telephone surveys) [39,45,48]. Geographically, five studies had health professionals responding Australia-wide [40,45,46,51,52]; two studies had representation from four states: New South Wales (NSW), Victoria (VIC), South Australia (SA), and Queensland (QLD) [32,33], one from SA and Northern Territory (NT) [21]; and seven were conducted specifically in NSW [20,38,42,43,47,48,53], four in VIC [35,37,39,46], three across QLD [40,44,54], two across SA [34,50], and one each in Tasmania (TAS) [45] and Western Australia (WA) [11]. Published between 1994 and 2022, sample sizes varied across these studies, from 22 [34] to a larger national sample of 51,840 [11] (Table 2). Only six studies reported gender-specific smoking prevalence [20,44,46,47,50,53]. Notably, assessing smoking prevalence was not the only outcome for the majority of these studies, they also explicitly explored other smoking-related domains like smoking habits and attitudes, attitudes towards providing smoking cessation care to patients [21,34,35,37,38,39,41,42,44], attitudes towards their smoking cessation [21,34,35,40,43,48,49,50,51,53], substance use [47] including alcohol consumption [11,32,33], or healthy or preventative lifestyle behaviors [41,45,47,53,54].

### 3.3. Diverse Range of Health Professionals

A varied range of Australian health professionals were considered in the included studies, notably physicians/medical officers including postgraduate trainee physicians, psychiatrists or general practitioners (GPs), dentists, optometrists, resident nurses, nursing staff or midwives, or mental health nurses or community nurses, and Aboriginal Health Workers (AHWs). These health professionals were also recruited from a variety of settings and institutions/associations across Australia and included the Royal Australian College of General Practitioners’ Family Medicine Program (FMP), GP practices, large hospitals, Dentists Associations, members of endorsed mental health nurses (EMHN), community centers in rural Australia, Aboriginal Primary Health Care Workers Forum (APHCWF), members of Optometry Australia, Aboriginal Community Controlled Health Services (ACCHs), the National Aboriginal and Torres Strait Islander Health Workers and Practitioners Association (NAATSIHWP), Nurses and Midwives Association, etc. (Table 2).

### 3.4. Measurement of Smoking

Nearly half of the studies (48%) did not explain how they defined current smoking status [20,21,33,34,35,38,39,40,43,44,45,52,54] (Table 4). Studies (*n* = 13) that defined current smoking also varied in how they measured the status. Some studies measured current smoking as daily smoking of cigarettes, pipes, or cigars [32,37,53], smoking at least 100 cigarettes in their lifetime, and currently smoking cigarettes, cigars, or pipes [42], smoking daily, weekly, or less often than weekly, smoking at least one cigarette per day [46], more than one cigarette/day, one cigar/week or chewing 30 g of tobacco for a month for at least the past year [41].

### 3.5. Smoking Prevalence Among Diverse Health Professionals

#### 3.5.1. Prevalence Among Physicians

There were seven studies under the purview of this scoping review (Figure 2) that dealt with smoking among physicians; these were published between 1995 and 2022, however, the actual data collection period varied considerably (Table 4). The postgraduate trainee physicians, psychiatrists, and general practitioners (*n* = 1361) working across NSW, QLD, VIC, and the SA state had a current smoking prevalence of 6% during 1990-91 [32]. Another study [33] was conducted in 1990 among medical practitioners enrolled in the Royal Australian College of General Practitioners’ Family Medicine Program (FMP) across the same four states. The FMP trainees (*n* = 908) had a current smoking prevalence of 4%, while a further 8.3 percent were ex-smokers. Randomly selected GPs from Victoria also had a current smoking prevalence of 4% in 1994 [35]. Medical staff (*n* = 185) at the Queen Elizabeth Hospital, Adelaide, showed a smoking prevalence of 3% in 1997 [34].

The longitudinal study based on the National Health Surveys (NHS) showed smoking prevalence between the years 1989–1990 and 2005 and had representation from all over Australia [11]. This revealed the chronological proportion of current smokers among Australian physicians at 10.2% in 1989–1990, 11.3% in 1995, and 10.6% in 2004–2005 (data were unavailable for physicians in the 2001 survey). Between 2004 and 2007, medical officers employed in four hospitals across SA and NT, namely Flinders Medical Centre (FMC), Alice Springs Hospital (ASH), Royal Adelaide Hospital (RAH), and the Queen Elizabeth Hospital (TQEH), were surveyed [21]; current smoking prevalence was observed at 12.1% at ASH in 2004, 6.6% at FMC in 2004, and 5.6% at RAH in 2005 and was markedly lower at 2% at TQEH in 2007, respectively.

Based on the Beyond Blue National Mental Health Survey, smoking status among specialty-trained doctors (vocational trainees, VT) working across all the Australian states (*n* = 1890) was measured in 2013 [36]. The overall prevalence of current smoking among VTs was found at 7.4%; however, there were variations across specialties, i.e., 6.6% among the obstetrics/gynecology VTs, 10.2% among emergency medicine/ICU VTs, 9.1% among anesthetic VTs, 6% among the internal medicine VTs, 10.7% among surgery VTs, and 11% among psychiatry VTs.

#### 3.5.2. Prevalence Among Dentists

Four studies measured smoking prevalence among Australian dentists during the defined timeframe (Figure 2). According to the 1993 study among Victorian dentists (*n* = 128) [37], the prevalence of daily smokers (of cigarettes, pipes, or cigars) was 6%, another 6% were non-daily smokers, and 29% were ex-smokers. Members of the Hunter Branch of the Australian Dental Association (ADA) of NSW and members of the Area Health Service Dental Service (*n* = 95) were surveyed in 1993 [38]. This showed only 3% of dentists as current smokers and 32% as ex-smokers. In 2001, 4% of the dentists (*n* = 250) working in Melbourne, Victoria [39] were current smokers. Members (*n* = 281) of the Australian Dental Association (ADA), Queensland Branch, had a current smoking prevalence of 3.9% in 2004, and 11% reported as former smokers [40].

#### 3.5.3. Prevalence Among Optometrists

This scoping review retrieved only one study [41] among Australian optometrists, which was carried out in 2013; a total of 283 members (56% female and 44% male) from Optometry Australia participated in that cross-sectional survey (Figure 2). Only 1% indicated being current smokers, with almost one in seven (13.3%) practitioners indicating having a prior smoking history.

#### 3.5.4. Prevalence Among Nurses

Eleven studies from this review dealt with smoking prevalence among Australian nurses and midwives, resident and enrolled nurses, mental health nurses, and community nurses (Figure 2). In 1991, the nursing staff (*n* = 335) of six large hospitals in the Hunter region of NSW state showed a current smoking prevalence of 21.7%, while 21.5% of the nursing staff were ex-smokers [42]. Nursing staff (*n* = 458) of the Queen Elizabeth Hospital, Adelaide, reported a smoking prevalence of 15.5% in 1997, which was rather higher than the prevalence found among the medical staff (3%) of the same hospital [34].

The Central Sydney Area Health Service (CSAHS) nursing staff (*n* = 1457) were also surveyed in 1997; this showed current smoking prevalence at 21% with 22% as ex-smokers [43]. The NHS [11] showed the chronological proportion of current smokers among Australian nurses at 29.1% in 1989–1990, 18% in 1995, 21.3% in 2001, and 18% in 2004–2005. The proportion of ex-smokers during the same period was 25%, 29.2%, 34.2%, and 27.2%, respectively. In 2005, surveyed mental health nurses (*n* = 289) from QLD reported 16% as current smokers [44]. The resident nurses employed in four hospitals across SA and the NT, namely Flinders Medical Centre (FMC), Alice Springs Hospital (ASH), Royal Adelaide Hospital (RAH), and the Queen Elizabeth Hospital (TQEH), had a current smoking prevalence of 21.3% at ASH in 2004, 19.1% at FMC in 2004, 6.1% at RAH in 2005, and 9.8% at TQEH in 2007, respectively [21]. In 2007, 11% of nurses in four public hospitals in VIC (*n* = 113) were reported as current smokers [46]. Five percent of the female community nurses working in four community centers in rural North West Tasmania were current smokers in 2010 [45]. Between 2011 and 2012, registered (RNs) and enrolled nurses (ENs) employed in two acute tertiary referral hospitals in metropolitan Sydney revealed 18% self-reporting as current smokers [47]. In 2014–15, members of the NSW Nurses and Midwives Association (*n* = 5041) showed 10.3% of the nurses and midwives as current smokers. The smoking rate among male nurses was 9.9% and 12.9% among female nurses [53]. Finally, nurses from regional QLD (*n* = 101) were investigated and showed a current smoking prevalence of 5.2% [54].

#### 3.5.5. Prevalence Among Aboriginal Healthcare Workers (AHWs)

This scoping review retrieved six studies relevant to smoking prevalence among AHWs in Australia (Figure 2). In 1995, the western NSW region AHWs (*n* = 22) had a very high current smoking prevalence of 63.6% [20]. This study also showed 40.9% of female and 22.7% of male AHWs as current smokers; additionally, 80% of the female AHWs smoked more than 20 cigarettes per day. AHWs across the Illawarra and Shoalhaven regions of NSW showed a current smoking prevalence of 71.4% in 2000, 54.8% in 2001, and 59.4% in 2002 [48]. Between 2006 and 2007, AHWs working in the Aboriginal Community Controlled Health Service (ACCHS) centers in WA revealed 31% as current smokers and another 31% as ex-smokers [49]. Smoking status and quitting smoking behavior of AHWs (*n* = 85) in SA were also explored and showed current smoking at 50.6%; female vs. male smoking was 55.4% and 41.4%, respectively, while 22.4% reported to be former smokers [50]. During 2012–13, among a national Australian sample of Aboriginal and Torres Strait Islander staff (*n* = 374) from the ACCHSs (2012–13), the current smoking rate was found to be 38%, while 24% were ex-smokers [51]. In 2021, the AHWs and Aboriginal Health Practitioners (AHPs) across Australia, who were members of NAATSIHWP, showed a much lower current smoking rate of 24.6% [52]

### 3.6. Predictors of Smoking

Twelve studies from this scoping review explored predictors of smoking among health professionals in Australia [11,32,33,34,36,37,40,42,43,46,47,50,51,53]. These are presented in three domains, notably biological and demographic predictors, psychological/physiological predictors, and environmental predictors (Table 3). Of all the predictors, a few of them were significantly associated with smoking in more than one study: they included male gender [46,53], having friends or family members who smoked [46,51], and specialty of the trainee physicians [32,46].

#### 3.6.1. Biological and Demographic Predictors

The study by Anna-Maree et al. [43] showed nurses aged 18–29 years and 30–39 years were more likely to be smokers than nurses with higher age groups. In contrast, a study by Lin et al. showed older nurses (age group 35–44 and 55–64) were more likely to be smokers than nurses aged between 25–34 (*p* < 0.05, respectively) [53]. Similarly, Derek et al.’s study [11] showed higher smoking prevalence to be related to older age (over 60 years); however, this association was not statistically significant. Studies conducted by Anna et al. [46] or Lin et al. [47] among nurses. The Lauran et al. [50] study conducted among AHWs also explored age as a possible predictor for smoking but could not elicit any significant association as such.

Like age, gender was also discussed as a possible predictor for smoking in eight of the included studies [32,33,34,36,40,43,46,47,50,53]. The Anna et al. [46] and Lin et al. study [53] showed male nurses were more likely to be smokers than female nurses (*p* < 0.05) [53]. The study by Rachel et al. [36] also showed gender as a predictor, as female specialty trainee doctors were found less likely to be smoking than their male counterparts (*p* < 0.05). However, studies conducted by Ann et al. [32] (among physicians, psychiatrists, and GPs), Ann et al. [33] (among medical practitioners), Ann-Maree et al. [43], Lin et al. [47] (among nurses), or Lauren et al. [50] (among the AHWs) could not find any association with gender (Table 5).

Marital status, having children, language spoken at home, geographic location, employment status, profession, specialty, place of training, year of graduation, and career length were the other possible demographic factors that were considered predictors of smoking in the included studies. A study by Anna et al. [46] showed single or separated/divorced nurses or nurses without children were more likely to be smoking (*p* < 0.05). Ann-Maree et al. [43] also showed nurses were more likely to be smokers if their main language spoken at home was English than nurses speaking in other languages (*p* < 0.05). Similarly, Amanda et al. [42] showed that place of training had a positive association, with hospital-trained nurses more likely to be smokers than university-trained nurses (*p* < 0.05). Rachel et al. [36] also showed that overseas-trained vocational trainee doctors were more likely to be smokers than the trainees from Australia (*p* < 0.05). Profession was also found to be a predictor in the Derek et al. study [11], as it showed nurses were more likely to be current smokers than physicians (OR 2.70, 95% CI 2.63–2.77). Anna et al. [46] explored area of specialty as a predictor and have shown that nurses working in psychiatry (*p* < 0.05), in emergency (*p* < 0.05), and in midwifery (*p* < 0.05) were more likely to be smokers than nurses employed in other specialties because of possible stress-related issues. Similarly, another study [32] in this review showed that the specialty of the trainee physicians group was a predictor, as trainee psychiatrists were found more likely to be current smokers than trainee physicians or trainee general practitioners (*p* < 0.05).

#### 3.6.2. Psychological/Psychosocial Predictors

The Lauren et al. study among South Australian AHWs showed that having lower emotional wellbeing was positively associated with being a current smoker; also, higher emotional wellbeing was observed among quitters (*p* < 0.05) and never-smokers (*p* < 0.05) [50].

#### 3.6.3. Environmental Predictors for Smoking

The influence/presence of other smokers was found to be a predictor of smoking in some of the selected studies. A study by Anna et al. [46] conducted among registered nurses showed parental smoking to be positively associated, as nurses who smoke were more likely to report smoking among their parents (*p* < 0.05); similar associations were observed among nurses having siblings who also smoked (*p* < 0.05), living with a smoker partner (*p* < 0.05), or living with a smoker friend (*p* < 0.05). Living with at least one smoker occupant in the households was found to be a predictor of smoking among the AHWs (*p* < 0.05) in one study [50]; however, the other study showed that ACCHS staff were less likely to be smokers than Indigenous communities when living with at least five closest family or friends who also smoked (AOR 0.56, 95% CI 0.34–0.04) [51].

### 3.7. Quality of the Studies

Of the 26 included studies, 25 could be regarded as good quality, and only one was scored to be of poor quality [55] (Appendix B).

## 4. Discussion

The prevalence of smoking among healthcare providers is a public health issue both for themselves and for their patients because they play a key role in combating the use of tobacco [56]. Even an earlier study conducted in a major hospital in Australia showed that non-smoking healthcare workers were more likely than smokers to see helping patients who wanted to quit smoking as definitely part of their role [57]. Overall, this review presents a picture of physicians, nurses, midwives, dentists, optometrists, and Aboriginal Health Workers (AHWs) in Australia in regard to their smoking prevalence and predictors of such behavior. Although this review explored all the literature available within this timeframe, except for one study conducted on AHWs, this review failed to identify recent literature that dealt with its objectives. This clearly indicates that smoking prevalence by healthcare workers has not been systematically reported in the literature [56] and thus identifies a huge gap in research.

Between 1990 and 2024, the prevalence of smoking among health professionals in Australia demonstrated varying declining trends. This review showed that for some of the health professional groups, this decline was in line with the national declining trend for the general (adult) population, while for others, this was not significantly proportionate.

Physicians in the current review were consistently less likely to be current smokers when compared with Australian nurses. However, there were contrasting smoking prevalence rates over the years; for example, the NNHS in 1989–90 showed a current smoking prevalence of 10.2% among Australian physicians [11]. There were studies, though, conducted around the same time in Australia that showed prevalence rates among physicians between 4% [33] and 6% [32]. These findings were consistent with studies conducted earlier, where 17% of US physicians were smoking in the 1980s [58], but this came down to only 4% by 1994 [59]. In New Zealand, only 3.4% of the medical staff smoked in 2006; studies conducted in the UK also showed similar consistently low smoking rates [60]. Smoking prevalence among physicians in the UK, US, and New Zealand has remained consistently lower ever since [10]. These rates among physicians are a contrast to some of the country rates in recent times like the Philippines (27.8%), Pakistan (29.5%), Turkey (31.6%), Argentina (20.1%), Iran (21.2%), Saudi Arabia (33.8%), and Spain (23.2%) [10,56]. A survey conducted in Italy in 2024 showed that 36.9% of the hospital doctors were current smokers [61]. By 2013, only 7.4% of the specialty-trained doctors were smoking in Australia, although this rate was higher among EDs (10.2%), surgeons (10.7%), or psychiatry (11.0%) VTs [36]. Unfortunately, we do not have an Australian study after 2013 conducted among physicians that could indicate whether this decline was valid at present. However, it can be assumed that smoking rates among Australian physicians were always lower than that of national adult smoking rates [62].

The important reasons behind the decline in smoking among physicians in countries including Australia could be attributable to stringent anti-smoking legislatures, tobacco control policies, and practices, including high taxes, plain packaging, and bans on advertisements in print and electronic media, to name a few. The other multifactorial reasons for the vast difference in smoking prevalence among physicians between countries could be due the cultural factors, marketing, lobbying of tobacco companies, and national health policies regarding tobacco control, as well as varying emphasis on the value of smoking cessation during basic medical and continuous professional education between European countries [63]. Some of the studies documented that an average of one-third of the smoking physicians continued to smoke because they failed to quit in previous quit attempts [64]. Apart from these factors, occupational stress, which is one of the key factors in addiction [6], or a country’s culture or wealth might have also influenced the smoking rates [56]. In countries with a downward trend in smoking among physicians over time, it may also be due to a cohort effect, with younger physicians less likely to start smoking than their older counterparts [65].

Of all the health professionals, except for AHWs, smoking prevalence among nursing staff in Australia has always been higher than that of physicians or dentists. In the 1970s, almost one in three Australian nurses smoked (32%) [66,67]. Over the years, smoking in this group also declined considerably. The highest smoking prevalence, as per this review, was 29.1% among the nurses in 1989–1990 [11]; since then, the decline continued to 21.3% in 2001 [11], 18% in 2004–2005 [11], a rate that persisted in 2011–12 [47], and then to the lowest prevalence of 10.3% during the 2014–15 period [53]. To understand this declining trend, the Australian adult smoking rate was 22.4% in 2001, 21.3% in 2004–05, 16.1% in 2011–12, and 14.5% in 2014–15 [62]; so, nurses were smoking more than adult Australians even until 2011–12. Like in Australia, declining trends among nursing staff were observed in other countries like the USA, Canada, and the UK. In the USA, the introduction of a national program to help nurses quit aided the progressive decline in smoking prevalence amongst nurses from 33.2% in 1976 to 8.4% in 2003 [46]; a similar decline was also observed in Canada from 32% in 1982 to 12% in 2002 [68] and in the UK, where it fell from 40% in 1984 to 20% in 1993 [69]. In New Zealand, the 2006 Census of Population and Dwellings showed midwifery and nursing professionals had a smoking prevalence of 13.6% [60]; however, this was lower in 2013, when 7.9% of female nurses and 9.2% of male nurses were found to be current smokers [70]. Before 1999, Japanese nurses had a smoking prevalence between 17.2% and 19.8%, but this came down to 8.2% among registered nurses and 4% among midwives during the 2013–14 period [71]. There are countries, however, with much higher smoking prevalence burden among nursing professionals like China (46.7% in 2012) [72] and Italy (36%) [73]. Like the physician group, we could not identify a more recent study (i.e., conducted in the last 5 years) focusing on Australian nurses to compare their latest smoking rates with those of the current Australian adult smoking rate of 10.7% [74] or those from around the globe. This would mean, at this juncture, there is insufficient comprehensive data about the smoking patterns of Australian nurses to draw valid conclusions regarding their current smoking status [44]. Like physicians, the reasons for the smoking decline among nursing professionals in Australia could be attributed to the sustained government tobacco control strategies, such as raising tobacco taxes, advertising bans, mass media public education campaigns, and comprehensive smoke-free environment legislation [75]. Moreover, the reduction in nurses’ smoking in Australia could be related to the change in nurse education to a tertiary degree, driving higher educational attainment. Across many countries, declining smoking among nurses may additionally mirror the decline in smoking among women as a whole, signifying the changing social norms of smoking among women or reflecting increasing public awareness of the harmful effects of tobacco use [76].

Dentists in Australia showed consistently lower smoking rates across the years; this overall low rate of tobacco usage, as revealed by this current and other reviews, suggests that dentists have one of the lowest smoking rates among all health professionals, much lower than the general population [18]. In Australia, this rate varied from 6% in 1994 [37] to 4% in 2003 [39] and 2005 [40]. For American dentists, it was 1% in 1994 [77] to 6% in 2005 [78]. Comparable smoking rates were also observed in the UK (5%) in 2010 and in Finland (3%) [79]. There were some notable exceptions such as in Italy, which showed a rather higher smoking prevalence of 33% among its dentists in 1997 [80], or 35% in Jordan in 2003 [81], or in Japan, where 28.9% of dentists were current smokers in the same year [82]. Apart from these few exceptions, reasons for a generally lower trend of tobacco smoking among dentists may be unclear; it could probably relate to their graphic awareness of the harmful effects tobacco consumption may incur for oral cavities [18,83] or be due in part to the incorporation of tougher legislative measures [83]. As in some countries like Poland, this could also be influenced by the feminization of the dental profession [84]. Additionally, like physicians, it was observed that dentists in most societies also tend to give up smoking before the general population [85].

Unlike other health professionals, the decrease in smoking prevalence is very slow among Indigenous people in Australia [86]. Such a decline was not obvious until recent times [20,52]. Historically, AHWs smoked more than the average Indigenous people; Bruce et al. [20] indicated that 63.6% of AHWs were current smokers in 1995, while 54.5% of Indigenous people were smoking nationally in 1994 [87]. This was about twice the proportion of Australians overall at that time (males 28%; females 22%) [88,89]. There were variable trends in smoking prevalence by AHWs in the subsequent years as well, i.e., 54.8% in 2001 [48] and 50.6% in 2013 [50]; these trends were somewhat consistent with national Indigenous male and female smoking prevalences of 52.6% and 47.4%, respectively, in 2008 [90]. These rates were also similar to the 2006 Aboriginal Peoples Survey (APS), which showed about 59% of Aboriginal Canadians smoked regularly [91]; these daily smoking rates were over three times that of all adults in Canada (17%) [92]. The highest decline among AHWS in Australia was shown by Michelle et al. during a 2021 survey, when 24.6% of AHWs and AHPs were found to be current smokers; this was lower than the 40.2% of Indigenous Australians who were still smoking nationally during 2018–19 [52].

The failure of a consistent decline among AHWs, until 2021, may be attributed to the high prevalence rate itself, which functions to normalize smoking in this population [50]. Other factors are stress emitting from racism and family and work expectations [48,51]; continually being in a smoking environment; the addictive nature of smoking; suboptimal understanding of nicotine dependence [49]; socioeconomic variables, including lower levels of perceived social support [50] or a lack of quitting support [48,51]; experiencing negative feelings, including loneliness, depression, or unhappiness [50]; or, interestingly, patients liking AHWs smoking with them, facilitating connections [93]. The recent most evidence, however, may augur a wind of change among the broader Aboriginal and Torres Strait Islander community in Australia that demonstrated successful reductions in smoking rates by promoting smoke-free behaviors [52]. This also strengthens and validates the continued targeted interventions for at-risk groups.

Smoking is a complex behavior that involves nicotine dependence, often guided by theories of smoking, especially the social cognitive theory or the health belief model, as this is affected by a host of factors related to smokers [21]. In this review, predictors that were found to be significantly related to higher smoking among health professionals (physicians, nurses, dentists, and AHWs) included age, gender, profession, marital status, smoking in the household/among peers/colleagues, language spoken at home, place of training, area of specialty, and lower emotional wellbeing.

Studies among health professionals have shown that early initiation of cigarette smoking has been associated with greater consumption, longer duration of smoking, and increased nicotine dependence [94,95,96]. Studies by Ann-Maree et al. and Lin et al. from this review showed younger age as a predictor [43,53]. The main reasons to have started smoking at an early age may be due to the influence of peer groups and friends, with smoking associated with a ‘high image’ among peers, staying away from families, or living with friends who also smoked [43]. However, in contrast to early age, older age was also found to be a predictor among nurses for higher smoking [53]; there are studies, also, where age was not a predictor [97,98].

Historical trends have shown gender as a predictor with males smoking more than females generally; this could be related to culture, masculinity, or social norms where peer pressure or self-esteem becomes a priority for the male gender [99]. Gender was also an important predictor of smoking in this review among nurses [46,53] and physicians [54]; this is corroborated by studies conducted among health professionals in Italy [97], Saudi Arabia [100], Armenia [101], Pakistan [2], Palestine [102], and Jordanian male physicians and nurses [103,104].

In contrast, there were studies, however, which demonstrated female health professionals with higher smoking rates [6]; this could be related to professions such as nursing, where predominantly more females are employed. Profession-wise, nurses were more likely to be smokers than physicians [11]; this finding was consistent with other studies [4,10]. Some studies on nurses’ smoking behavior demonstrated that smoking is a coping mechanism against stress, caused by peer behavior and the nursing environment [10]. Like profession, the specialty of the health professionals (like trainee psychiatrists smoking more than trainee physicians or trainee general practitioners) [32,46] was substantiated by studies that showed trainee psychiatrists [5,32], psychiatry residents [5,32], nurses, respiratory physicians, and occupational therapists smoking more than physicians or other categories of health professionals [4]. This was also evident in a few other studies [5,105], where surgeons or health professionals working in operating rooms or ICUs were shown to be likely smokers [104]. The reasons for these observations are not clear but could be attributable to stress, personal characteristics, and/or professional role conflicts [106].

Being single or separated/divorced, parental smoking, or living with a smoking partner or siblings were also identified as predictors [46]; these were consistent with studies conducted elsewhere [102] that showed higher smoking among close friends and colleagues [2,51] or where there was a positive family history of smoking [102].

Where health professionals received their training was also recognized as a predictor with hospital-trained nurses more likely to smoke than university-trained nurses [42]; similar findings were observed elsewhere where physicians and nurses from public health institutes and general hospitals [107] or health professionals from public hospitals were smoking more than private hospitals. This could be attributed to stricter anti-tobacco policies or their implementation as such in private institutions [2].

Rural residents are likely to smoke more due to certain demographic characteristics, such as low income, low educational attainment, and lack of health coverage [4]. However, rural vs. urban residence was found not to be a predictor in this review, as was consistent with some studies in the USA [4] and elsewhere [102].

### Limitations of the Review

There were some limitations to this scoping review process. There could be potential inherent publication or selection bias due to the inclusion and exclusion criteria. This review only considered articles that were published, as well as articles written in English only—essentially exposing this review to publication and selection bias. A majority of the included studies were also cross-sectional and anonymous, and only a few longitudinal data sets (among health professionals) were available and hence were subjected to self-reporting; this might mean more non-smokers completed the questionnaires, resulting in underestimation and underreporting of the prevalence in some of the studies. So, when we are observing a strong declining trend in smoking rates in Australia, one must take into account the potential data biases (i.e., the impact of self-reported biases) to balance the interpretation. Also, by nature, a cross-sectional design is unlikely to capture change in smoking behavior, since smoking behavior is measured only at one point in time, so such a study will yield heterogeneity among smokers who belong to the same stage of smoking [108]. There was also a lack of statistical testing for trends and predictors in many of the reviewed studies. The geographical and temporal representation of the included studies in this review was also a challenge.

One factor across most of the studies, however, was a lack of standardization regarding the definition of ‘current smoking’. Methodological sources of discrepancy due to differences in survey questions used to define current smoking are nothing new in research though [109]. Nearly half of the studies did not mention a consistent definition of ‘current smoking’, which may affect the reliability of aggregated findings [108]. Most studies often use different definitions of smoking, which has often made it difficult to compare findings across studies. The variations in definitions include the number of cigarettes/pipes/cigars smoked per day, the number of days smoked per week or month, or the amount of lifetime cigarette use; all have served as a proxy for smoking prevalence. For example, in the US, the National Health Interview Survey (NHIS) limits its question about current smoking to respondents who reported smoking ≥100 cigarettes in their lifetime; however, the National Survey on Drug Use and Health (NSDUH) does not designate a cut-point for number of lifetime cigarettes smoked [109]. This invites potential discrepancies across the two data sets. There are other implications for both the self-reporting nature of the studies included in this review and the varying definitions of current smoking. These may potentially indicate that there has been an underestimation of smoking prevalence among healthcare workers or that the accuracy of the smoking trends of Australian healthcare workers is compromised and any comparisons made need to be interpreted with caution.

A majority of the studies also did not provide gender-specific prevalence rates, and not all the studies discussed predictors of smoking as such. A smaller sample size, acceptable margin of standard error, and varying response rates could also add complexity in terms of the generalizability of the findings across some Australian states or the country as a whole.

## 5. Conclusions

This scoping review was able to provide a critical examination of the literature on smoking prevalence and its predictors among healthcare professionals in Australia. To our knowledge, this review was the first on this topic. Hence, in essence, this is unique, as it contributes to the current literature that attempted to collate information on the prevalence and predictors of smoking across a diverse range of healthcare professional groups in Australia that can affect tobacco-related morbidity and mortality among their patients and in the community. Despite being aware of the deleterious effects of smoking, and even though they serve as role models for the general population, this review showed that Australian healthcare professionals continue to smoke. There is a declining trend observed though, which is encouraging and may have a potential impact on broader public health policies. However, smoking by nurses and AHWs continues to be a concern. This underscores the need to further the anti-smoking programs within healthcare settings. As evidenced from this review, predictors like male gender in nurses, specialty of the physicians and, generally, older age of nurses and midwives (that were associated with higher smoking rates) need to be taken into account to tailor smoking cessation interventions for them. These may include, but are not limited to, having social media influencers/champions of change for smoking cessation in health industries, smoking cessation counseling specifically meant for them, more pronounced roles from the health practitioners’ regulatory agencies, etc. AHWs are a distinct subgroup as identified in this review and also a priority group for interventions. The insights from the latest study [52] that showed a reasonable decline in AHWs’ smoking status may contribute to practical policy recommendations, including targeted interventions for this high-risk group. This may also include, among others, cultural competency, engaging with the Aboriginal community and its leaders, and factoring in how the community perceives smoking cessation interventions directed towards them and the AHWs. This review also highlighted that evidence on smoking among healthcare professionals is mostly based on cross-sectional studies, which may not represent the wide spectrum of this workforce across the geographical landscape of Australia. Representative regional or national level data is a key gap, and the lack of homogeneity of the data is another gap. The paucity of recent literature, as revealed by this scoping review, also highlights the need for more up-to-date information to further understand the declining smoking trends among the Australian health workforce.

## 6. Future Directions

Research focusing on the smoking behavior of healthcare professionals in Australia is limited, although most of the frontline healthcare professionals have the opportunity to offer smoking cessation services to the community. This review highlighted the relative lack of more up-to-date data; there was no information available after 2015 in regard to physicians’, nurses’, dentists’, or other healthcare workers’ smoking habits. This review also demonstrated a disproportionate decline in smoking prevalence rates among the health workforce spectrum in Australia, with nurses and Aboriginal health workers still demonstrating higher smoking rates. Since their own smoking behavior can compromise the smoking cessation support provided to their patients, further research into healthcare professionals’ smoking behavior across the Australian states and territories is warranted. To further aid in prioritizing policies and smoking cessation guidelines, researchers need to use representative samples using homogenous definitions for current smoking so that smoking trends can be examined more accurately. More research is also required to understand the dynamics of the smoking decline among the priority AHW groups and nurses. Geographic determinants like urban vs. rural demographics also need to be explored in relation to smoking behavior by healthcare workers to examine the potential differences between the two. All of these will not only increase our knowledge base and address gaps in the literature but will also help us to use that evidence where it is most needed in regard to tobacco control cessation policy, strategies, and practices.

## Figures and Tables

**Figure 1 ijerph-22-00113-f001:**
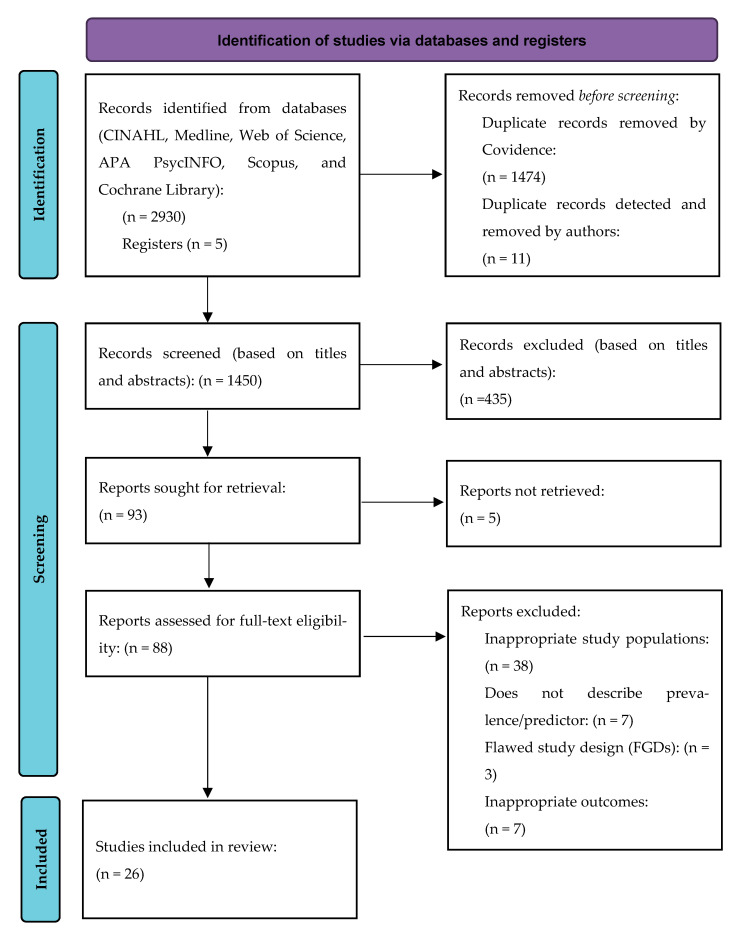
Preferred Reporting Item for Systematic Reviews and Meta-Analysis for Scoping Review (PRISMA-ScR) flow diagram. Twenty-six studies met the inclusion criteria of the ScR.

**Figure 2 ijerph-22-00113-f002:**
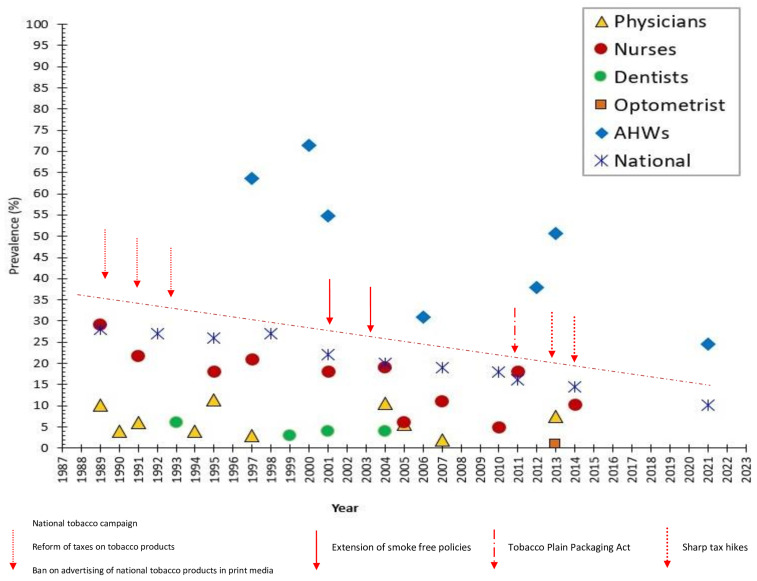
Trends in smoking among different healthcare professionals in Australia.

**Table 1 ijerph-22-00113-t001:** Inclusion and Exclusion criteria of the scoping review (PCC).

	Criterion	Inclusion	Exclusion
Population	Sample	Health professionals living and working in Australia. They could be medical doctors, general practitioners (GP), doctors working at hospitals, specialists, dentists, optometrists, licensed nurse practitioners, practice nurses, registered nurses, mental health support workers, pharmacists, physician assistants, or Aboriginal and Torres Strait Islander health practitioners.	Studies involving smoking among students enrolled in healthcare professions or allied health service providers, including administrative staff in GP or hospital settings, technicians, therapists (physical, occupational, or speech), chiropractors, or social workers because they are not directly involved in tobacco cessation advice.
	Geographical place of study	Australia	Studies conducted elsewhere.
	Setting	No setting (such as hospitals, including teaching or tertiary hospitals, GP clinics, dental practices, primary health care, community clinics or services, etc.) recognized as long as the study focused on smoking, prevalence of smoking, or predictors of smoking among health professionals.	Nil
Concept	Study focus	Studies that focus on smoking, prevalence of smoking, or predictors of smoking among health professionals.	Studies that do not focus on smoking, prevalence of smoking, or predictors of smoking among health professionals, as well as publications that deal with smokeless tobacco, including chewing tobacco, dry snuff, moist sniff, or passive smoking.
	Time period	1990–2023, this was set to identify trends and developments over the last 2 decades.	Published before 1990.
Context	Type of article	Original research, review articles, reports, or case studies published in academic, peer-reviewed, or scholarly journals, books, reports, or fact sheets. Also systemic reviews or gray literature.	Conference abstracts, letters to editors, or unpublished works.
	Language	English	Non-English

**Table 2 ijerph-22-00113-t002:** Studies on prevalence of smoking among healthcare professionals in Australia.

Specialty	Study	Current Smoking Prevalence (%) Based on Definition of Smoking				Sample Size	Study Sites
		Smoking daily	Smoking daily to weekly	Smoking in last 1 yr	Not defined		
Physicians	Roche et al. (1995) [32]	6				1361	NSW, QLD, VIC, SA
	Roche, Parle and Saunders (1996) [33]	4				908	NSW, QLD, VIC, SA
	Jones, Crocker and Ruffin (1998) [34]				3	185	SA
	McCall, Maher and Piterman (1999) [35]				4	318	VIC
	Smith (2007) [11]				10.2 (1989–1990), 11.3 (1995), 10.6 (2004–2005)	51,840 (1989–90), 49,680 (1995), 23,400 (2004–2005)	Australia
	Jones and Williams (2010) [21]				2–12.1 *	4606 **	SA, NT
	Wong et al. (2022) [36]				7.4	1890	Australia
Dentists	Mullins (1994) [37]			6		128	VIC
	Clover et al. (1999) [38],				3	95	NSW
	Trotter and Worcester (2003) [39]				4	250	VIC
	Smith and Leggat (2005) [40]				4	281	QLD
Optometrists	Downie and Keller (2015) [41]	1				283	Australia
Nurses	Jones, Crocker and Ruffin (1998) [34]		15.5			458	SA
	Nagle, Schofield and Redman (1999) [42]				21.7	335	NSW
	Huges and Rissel (1999) [43]		21			1457	NSW
	Smith (2007) [11]				29.1 (1989–1990), 18.0 (1995), 21.3 (2001), 18.0 (2004–2005)	51,840 (1989–90), 49,680 (1995), 24,840 (2001), 23,400 (2004–2005)	Australia
	Jones and Williams (2010) [21]				6.1–21.3 ***	4490 ****	SA, NT
	Dwyer, Bradshaw and Happell (2009) [44]				16	289	QLD
	Newman and Berens (2010) [45]	5				40	TAS
	Berkelmans et al. (2011) [46]	11				1029	VIC
	Perry, Gallagher and Duffield (2015) [47]			18		381	NSW
Aboriginal Health Workers	Andrews, Oates and Naden (1997) [20]				63.4	22	NSW
	Mark et al. (2005) [48]				71.4	98	NSW
	Pilkington et al. (2009) [49]	31				36	WA
	Maksimovic et al. (2013) [50]				50.6	85	SA
	Thomas et al. (2015) [51]			38		374	Australia
	Kennedy et al. (2023) [52]				24.6	256	Australia

* Prevalence at individual hospital: 2.1 (2004, *ASH) 6.6 (2004, *FMC) 5.6 (2005, *RAH) 2 (2007, *TQEH). ** Sum of sample across 4 hospitals: TQEH: 1298; RAH: 1565; ASH: 283; FMC: 1460 (ASH: Alice Springs Hospital; FMC: Flinders Medical Centre; TQEH: The Queen Elizabeth Hospital; RAH: Royal Adelaide Hospital). *** Prevalence at individual hospital: 21.3 (2004, *ASH) 19.1 (2004, *FMC) 6.1 (2005, *RAH) 9.8 (2007, *TQEH). **** Sample across 4 hospitals: TQEH: 924; RAH: 1165; ASH: 269; FMC: 2132.

**Table 3 ijerph-22-00113-t003:** Studies on Predictors of smoking among healthcare professionals in Australia.

Predictor		Power of Association	Specialty	Study
Gender	Male	*p* < 0.01	Nurses	Berkelmans et al. (2011) [46]
		*p* < 0.001	Nurses and Midwives	Perry et al. (2018) [53]
		*p* < 0.001	Specialty Trainee Doctors	Wong et al. (2022) [36]
Age	18–29 yrs	*p* < 0.001, AOR: 3.43, 95% CI: 1.46–8.05	Nurses	Huges and Rissel (1999) [43]
	30–39 yrs	*p* < 0.001, AOR: 3.31, 95% CI: 1.44–7.63		
	35–44 yrs	*p* < 0.05, OR: 1.62, 95% CI: 1.09 –2.42	Nurses and Midwives	Perry et al. (2018) [53]
	55–64 yrs	*p* < 0.05, OR: 1.54, 95% CI: 1.06–2.23		
Language spoken at home	English	*p* < 0.05, AOR: 0.32, 95% CI: 0.15–0.70	Nurses	Huges and Rissel (1999) [43]
Marital status	Single or separated/divorced	*p* < 0.01	Nurses	Berkelmans et al. (2011) [46]
Having no children		*p* < 0.01	Nurses	Berkelmans et al. (2011) [46]
Parental smoking		*p* < 0.01	Nurses	Berkelmans et al. (2011) [46]
Smoking by siblings		*p* < 0.01	Nurses	Berkelmans et al. (2011) [46]
Living with a partner/another who smokes		*p* < 0.01	Nurses	Berkelmans et al. (2011) [46]
		*p* < 0.01	Aboriginal Health Workers	Maksimovic et al. (2013) [50]
Have friends/family who smoke		*p* < 0.01	Nurses	Berkelmans et al. (2011) [46]
		AOR: 0.56, 95% CI: 0.34–0.94	Aboriginal and Torres Strait Islander Health Service (ACCHS) Staff	Thomas et al. (2015) [51]
Profession	Nurses more likely than physicians		Physicians and Nurses	Smith (2007) [11]
	2.70 times more likely in 1989–90	OR 2.70, 95% CI 2.63–2.77		
	1.61 times more in 1995	OR 1.61, 95% CI 1.57–1.66		
	1.71 times more in 2005	OR 1.71, 95% CI 1.67–1.75		
Specialty	Trainee psychiatrists (11%) more likely than trainee physicians (5%) or trainee GPs (4%)	*p* < 0.001	Physicians	Roche et al. (1995) [32]
	Working in psychiatry	*p* < 0.01	Physicians	Berkelmans et al. (2011) [46]
	Working in emergency department	*p* < 0.01	Physicians	Berkelmans et al. (2011) [46]
Place of training	Overseas trained more likely (9.6%) than Australian born trainees (6.1%)	*p* < 0.05	Specialty Trainee Doctors	Wong et al. (2022) [36]
	Hospital trained more likely than university trained	*p* < 0.05	Nurses	Nagle, Schofield and Redman (1999) [42]

**Table 4 ijerph-22-00113-t004:** Specific definitions of current smoking and critique of the included studies.

Study	Specialty	Current Smoking Definition	Comments
Mullins (1994) [37]	Dentists	Daily smoker of cigarettes, pipes, or cigars.	Small sample size, findings may not be generalizable.
Roche et al. (1995) [32]	Post graduate trainee physicians, psychiatrists, and general practitioners	Light smoker: 1–5 cigarettes/day; medium smoker: 6–20 cigarettes/day; heavy smoker: >20 cigarettes/day	Response rate of 55% is satisfactory for a mail survey, however, attitudes of non-responders in terms of their smoking habits not known.
Roche, Parle and Saunders (1996) [33]	Trainee medical practitioners	Not defined	Smoking attitudes of non-respondents remain unknown.
Andrews, Oates and Naden (1997) [20]	Aboriginal Health Workers (AHWs)	Not defined	Very small sample size, findings may not be generalizable.
Jones, Crocker and Ruffin (1998) [34]	Medical and Nursing staff	Not defined	Return of questionnaire was voluntary, which may underestimate the actual smoking prevalence as there was a greater likelihood that smokers would not return the questionnaire.
McCall, Maher and Piterman (1999) [35]	General practitioners	Not defined	Response rate of 58.5% was satisfactory, however, response bias may have occurred due to non-smoking physicians being more likely to return questionnaire, leading to underestimation.
Clover et al. (1999) [38]	Dentists	Not defined	Although response rate was satisfactory, responses were based on self-reported behavior, so reporting bias may have been an issue.
Nagle, Schofield and Redman (1999) [42]	Nursing staff	Smoked at least 100 cigarettes in their life and currently smoking cigarettes, cigars or pipes (in the last 4 days).	Sample only representative of the nursing staff of Hunter region.
Huges and Rissel (1999) [43]	Nurses	Not defined	A large sample size, although sampling frame for nurses not known.
Trotter and Worcester (2003) [39]	Dentists	Not defined	Sample may not be representative across Victoria, also more proactive dentist’s may have participated.
Smith and Leggat (2005) [40]	Dentists	Not defined	Possibility of selection bias as current smokers may be unwilling to return questionnaire; there could be underestimation.
Mark et al. (2005) [48]	Aboriginal Health Workers (AHWs)	Smoking regularly, more than one pack a day or occasionally but not every day.	Self-reported, so may be associated with reporting bias.
Smith (2007) [11]	Physicians and Nurses	Smoking daily, weekly or less often than weekly.	Very large sample in all the four surveys that included physicians and nurses. Use of trained interviewers instead of postal surveys may have been more appealing to the smoking demographic. The physician’s sample had a standard error of 50% in the 1989–90 data, as well as 25–50% standard error in 2004–2005 data.
Dwyer, Bradshaw and Happell (2009) [44]	Mental Health Nurse	Not defined	Low response rate may not be representative of the EMHNs, not possible to explore how the non-responders differed systematically from the sample.
Pilkington et al. (2009) [49]	Aboriginal Health Workers	Smoking at least one cigarette per day or seven per week.	Low recruitment rate may compromise the representativeness of the sample; there could be sample bias as well.
Jones and Williams (2010) [21]	Medical officers	Not defined	Lowest prevalence at TQEH could be attributed to a supportive program. Sample size and response rate inclusive of all staffs across these 4 hospitals. Also, given the nature of the surveys being self-reported, possibility of less self-identification as smokers.
	Resident Nurse	Not defined	
Newman and Berens (2010) [45]	Community Nurses	Not defined	Although response rate was high, sample size was not large enough to ensure findings are generalizable.
Berkelmans et al. (2011) [46]	Nurses	Regular smoker of at least one cigarette per day.	Response rate did not allow for generalizability of the results, self-selection and self-report of smoking status allowed potential sources of respondent bias or underestimation of true smoking prevalence rate.
Maksimovic et al. (2013) [50]	Aboriginal Health Workers (AHWs)	Currently smoking, smoking status assessed on a nominal scale.	Relatively small sample size; results may not be generalizable.
Downie and Keller (2015) [41]	Optometrists	A person who smokes more than one cigarette/day, 1 cigar/week or chews 30 g of chewing tobacco for a month, for at least the past year.	Very low response rate, potential for selection bias; also self-reported, so concerns regarding truthfulness of responses.
Thomas (2015) [51]	Aboriginal and Torres Strait Islander health service (ACCHS) staff	Not defined	The sample included ACCHS staff, not just AHWs, so results need to be interpreted with caution as role of ACCHS can vary across country.
Perry, Gallagher and Duffield (2015) [47]	Registered Nurses and Enrolled Nurses	Not defined	Study findings may not be representative of NSW nursing population; also data were self-reported, so possible the could be underestimation.
Perry et al. (2018) [53]	Nurses and Midwives	Somebody who smokes daily.	Self-selected and self-reported survey, so potential challenges to response veracity.
Heidke, madsen and Langham (2020) [54]	Nurses	Not defined	Current smoking criteria not clearly stated; study used convenience sample in a single regional area, so results are not representative.
Wong et al. (2022) [36]	Specialty trainee doctors	Current smoking prevalence was expressed as consumption of tobacco at least once every 6 months.	Low response rates, self-reported data may have led to non-response or reporting bias, as substance use can potentially be influenced by factors such as fear of stigmatization. Current prevalence of tobacco determined as at least once every 6 months may have resulted in overestimation. Data were collected in 2013, so may not represent current use.
Kennedy et al. (2023) [52]	Aboriginal Health Workers and Aboriginal Health Practitioners	Not defined	Low response rate, self-reported bias; there could be differences in terms of training and skills between the AHWs and AHPs, which may have resulted in lower prevalence rates.

**Table 5 ijerph-22-00113-t005:** Studies on Predictors of smoking among healthcare professionals in Australia (explored and found non-significant).

Predictor		Power of Association	Specialty	Study
Gender	Male	*p* > 0.05	Physicians	Roche et al. (1995) [32]
		*p* > 0.05	Physicians	Roche, Parle and Saunders (1996) [33]
		*p* > 0.05	Physicians and Nurses	
		AOR 1.51, 95% CI: 0.89–2.39	Nurses	Huges and Rissel (1999) [43]
		*p* > 0.05	Aboriginal Health Workers	Maksimovic (2013) [50]
		*p* > 0.05	Nurses	Dwyer, Bradshaw and Happell (2009) [44]
Age	>60 years	*p* > 0.05	Dentists	Smith and Leggat (2005) [40]
		*p* > 0.05	Nurses	Berkelmans et al. (2011) [46]
		*p* > 0.05	Aboriginal Health Workers	Maksimovic (2013) [50]
Year of graduation		95% CI: 5.08–19.08	Dentists	Mullins, (1994) [37]
Working hours	Between 25 and 35 h/week	Not assessed	Dentists	Smith and Leggat (2005) [40]
Career length	Worked > 40 years	Not assessed	Dentists	Smith and Leggat (2005) [40]
Specialty		*p* > 0.05	Specialty Trainee Doctors	Wong et al. (2022) [36]
Categories of nursing positions	Between registered nurses, clinical nurse consultants, nursing unit managers, and enrolled nurses	*p* > 0.05	Nurses	Nagle, Schofield and Redman (1999) [42]
Geographic locations		*p* > 0.05	Aboriginal Health Workers	Maksimovic et al. (2013) [50]
Employment status		*p* > 0.05	Aboriginal Health Workers	Maksimovic et al. (2013) [50]
Number of children living with AHW		*p* > 0.05	Aboriginal Health Workers	Maksimovic et al. (2013) [50]
Number of people in the household		*p* > 0.05	Aboriginal Health Workers	Maksimovic et al. (2013) [50]
Number of people supported on their wage		*p* > 0.05	Aboriginal Health Workers	Maksimovic et al. (2013) [50]

## Data Availability

Not applicable.

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
