# Peer review of "Smoking Among Healthcare Professionals in Australia: A Scoping Review"

_ijerph, 2025, doi:10.3390/ijerph22010113_

Round 1

Reviewer 1 Report

Comments and Suggestions for Authors

The author should make an effort to address my comments in order to improve the quality of the paper. I have attached the details.

Author Response

Thank you so much for reviewing our scoping review and providing a thorough and constructive feedback. We highly appreciate this. We have tried to address your comments to the best of our ability and we think this has significantly improved our revised manuscript. Please find below the reply to specific comments made:

  1. Title

The title is very narrow. However, considering the nature of a scoping review, the topic should be a little bit broader in order to capture adequate information in the literature. I hope this can be improved.

Response:

Thanks for your valuable feedback. As per suggestion, we have brought a change in the title to make the review a bit broader. Previously the title was, ‘Prevalence and predictors of smoking among healthcare professionals in Australia: A scoping review’. The revised title now is ‘Smoking among healthcare professionals in Australia: A scoping review’ [Line: 2].

  1. Introduction

Despite the topic of this manuscript being too narrow, the authors do attempt to demonstrate the need for a scoping review in the final sections of the introduction, which is appreciated. However, in order to comprehensively document the information related to the topic, it would be better if the authors revise the title and expand the content in the body of the introduction section. This is because the review is overly simplified.

Additionally, the research question of this scoping review should be separately presented.

Response:

We totally agree with your comment in this section. The content of the body of the introduction has been almost entirely revised now which captures not only the broader title but also logically documented the sequence of the paragraphs linking the objectives of the review in the end.

A research question to this review has now been added and presented in the revised manuscript [Line: 28 - 112].

  1. Method section

It would be helpful if the author(s) of this work documented both the data extraction tool and the software they used. The search terms could also be improved, as you only used prevalence, and predictors of smoking.' Do you think other researchers would report the same results if these search terms were used? You should also document the PCC (Participant, Concept, Context) framework for the eligibility of studies, as this is very critical. Also, I recommend that the protocol be registered on OSF (Open Science Framework).

Response:

Thanks for your feedback. As mentioned in the manuscript, we have used endnote and covidence software tools to streamline the data review process and all three of the researchers were involved in every stages of screening. In the revised manuscript we have added the use of a standardized data extraction and quality assessment criteria form that we have used and also the percentage agreement to ensure inter-rater reliability and resolving disagreements. We have mentioned about the use of PCC framework to construct the search terms and also to construct the research question. We did not register our review protocol in Prospero or in OSF.  Since our review is already completed, we will not consider the registration at this stage. However, this is a crucial advice which we are taking on board for our upcoming research works [Line:  200-210] 

  1. Results

 The reasons for exclusion should be elaborated for outsiders.

Response:

Thanks for your feedback. Accordingly, reasons for exclusions have been elaborated in the revised manuscript now [P: ]

.

Too many subheadings are misleading. Thus, the following statements should be improved:

2 3.4.1 Prevalence among physicians

3.4.2 Prevalence among dentists

3.4.3 Prevalence among optometrists

3.4.4 Prevalence among nurses

3.4.5 Prevalence among Aboriginal Healthcare Workers (AHWs).

Response:

Thanks for your valuable feedback. The uniqueness of the scoping review is that it captured smoking prevalence and predictor related information from a wide array of healthcare professionals in Australia. We did not know before from how many groups of healthcare workers we will be seeing a literature or two in regard to our objectives. In the end, we had these 5 groups of workers and that is why thought of describing the individual studies under each of these sub-headings. Omitting them might be confusing as there are 26 studies across these 5 groups of health professionals [Line: 344 - 405].

  1. Discussion

In order to improve the quality of the manuscript, the authors should be expected to thoroughly revise the discussion section by considering the following critical points:

  1. Compare the findings with those from comparable populations and settings.
  2. Provide possible explanations for any disparities between this study's findings and prior literature, after carefully reviewing related studies.
  3. What are the applications of your research findings? For example, explain how your findings contribute to the general understanding of the topic, extending beyond what others have discovered, and provide examples of why this increased understanding is important.
  4. State how you relate your study to existing theories in the literature. You should be familiar with the established theories on the subject matter and link your findings to them (whether they agree or not).
  5. Indicate the implications of these findings.

Response:

Thank you so much for your important feedback. We have mostly acknowledged your comments and have worked on these points to update the revised manuscript. However, please note the greatest challenge the authors faced to compare and contrast with contemporary data was not feasible because of paucity of data, not only from Australian context in the last 6-7 years but from the global stage as well. Even the few systematic reviews and meta-analyses that were carried out in recent times, data collection period for the included studies, be it for physicians or nurses or dentists or other allied health workers, were only up to 2012-2013 maximum. This has significantly reduced the comparable populations and settings discussion. We have tried to touch upon the disparities of the study findings also looked for possible causes for decline or a lack of decline in Australia and elsewhere. We have significantly modified the conclusion section to accommodate the applications of our research findings. We have briefly touched upon the theories in relation to smoking when we discussed about the predictors [Line: 483 - 778].

Strengthens and Limitations should be separately documented

Response:

It is documented separately now [Line: 718 - 734].

  1. Conclusions

Conclusions should be logical, clearly explained, and consider any limitations of the data or analysis.

  • Recommendations: Recommendations are specific actions that can be taken based on the findings and conclusions.

Therefore, make sound, actionable, conclusion-based recommendations after revising the conclusion section. The recommendations should address policy, practice, and further research, as you tried to say.

Response:

Thank you for your valuable feedback. Conclusion section has been revised as per your suggestion. Recommendations and conclusion read more sound and actionable now [Line: 759 - 778].

  1. References

Check thoroughly!

Response:

Done [P: 35-40]

Reviewer 2 Report

Comments and Suggestions for Authors

―    The introduction establishes the premise that healthcare professionals who smoke are less likely to provide effective smoking cessation advice (Lines 1–13). This provides a logical basis for the study. It discusses the influence of smoking habits among healthcare professionals on public perception and health promotion roles, which aligns with the study's relevance to public health (Lines 15–20). The argument is diluted by general statements about healthcare professionals' responsibilities (e.g., lines 12–20). For example, the relationship between declining smoking rates and the remaining prevalence among specific healthcare groups is not articulated clearly enough. The authors do not sufficiently connect this data to previous studies in similar settings or regions. Including comparisons to international contexts and cite current data (within the past 5–7 years), such as updated latest global or national reports on smoking behaviours among healthcare workers would improve the introduction. The introduction does not clearly explain how the study differs from or improves past research. Clearly write a central argument summarizing the need to study smoking prevalence and predictors among healthcare professionals, emphasizing its relevance to Australia’s unique healthcare environment. Focus on the unique aspects of the argument (e.g., smoking prevalence trends and their impact on healthcare efficacy, recent public health campaigns, technological advancements in data collection, or shifting societal attitudes toward smoking in healthcare.
―    Lines 30–40: condense the repetitive details about the importance of healthcare professionals in smoking cessation.
―    The paper emphasizes the innovative approach to studying smoking prevalence and predictors among Aboriginal Health Workers (AHWs), a subgroup often overlooked in research (Lines 47–60).  The study’s use of PRISMA and scoping review methodologies implies a novel methodological perspective.  But the originality of the analysed predictors is not underlined. So, emphasize the novel aspects of the study, such as its comprehensive scope, inclusion of underrepresented groups like AHWs (specific challenges they face regarding smoking cessation or the cultural context of smoking within Aboriginal communities), or innovative methodological approaches. Also connect the originality to actionable outcomes, such as informing tailored interventions or policy changes.
―    The claim that smoking prevalence data among Australian healthcare professionals is sparse (lines 41–50) is not convincingly supported by examples.
―    The aims, focusing on prevalence and predictors, are clearly stated at the end of the introduction (lines 61–64). However, their specificity feels unexpected, as the introduction does not clearly lead to these objectives; linking the discussion of gaps more directly to the aims would create a smoother transition.
―    While the methodology is well-aligned with scoping reviews, the rationale for choosing this method over others (e.g., systematic review) is not explicitly stated.
―    Can you provide a detailed justification for the selection of databases included in the literature search, particularly in relation to their relevance to the interdisciplinary nature of the research topic, which spans public health, psychology, and healthcare practices?
―    The explanation of the screening and selection processes lacks sufficient detail. Although Covidence is recognized as a resource to simplify the procedure, providing more information on how inter-rater reliability was assessed (e.g., percentage agreement, kappa scores) would improve the study's transparency (Lines 125–133).
―    The confidence on self-reported data from the included studies introduces a risk of reporting bias. Nearly half of the studies did not mention a consistent definition of “current smoking status” (Lines 196–203), which may affect the reliability of aggregated findings.
―    Provide a more detailed discussion of potential limitations, particularly those related to selection bias, reporting bias, and geographic representation within the included studies.
―    While data extraction is mentioned, the process is not described in detail (Lines 124–133). For example, what specific variables were charted, and how were they categorized (e.g., demographic, psychological, environmental)? Clarify how data were extracted and whether a standardized coding scheme was applied to classify prevalence, predictors, or healthcare groups. If no statistical analysis was conducted, clearly state this to avoid confusion.
―    Reporting of search terms and the Boolean operators used in each database search is not sufficiently detailed. Provide a complete list of search strategies used in each database, including exact terms, truncations, and Boolean combinations, either in the text or as supplementary material.
―    Figure 1 lacks detail on reasons for exclusion (e.g., how many papers were excluded due to non-relevance vs. insufficient data). Adding more specific categories would strengthen its utility.
―    Include annotations in Figure 2 (Smoking Trends) to highlight key patterns significant deviations, linking them to relevant societal, policy, or demographic factors.
―    Table 2 (Prevalence of Smoking) and Table 4 (Study Characteristics) are too detailed and difficult to navigate. For instance, Table 2 includes excessive descriptive text that could be simplified.
―    In Table 2, the smoking definitions vary across studies (e.g., daily smokers, occasional smokers), but these inconsistencies are not visually highlighted. This reduces the comparability of the presented data. Clearly define smoking statuses across all studies and consider using symbols or color codes to visually highlight variations.
―    Table 3 (Predictors of Smoking) lists predictors but does not visually rank or quantify their significance, making it harder to assess which factors are significant. No statistical weight or ranking for the predictors is provided. For example, which predictor had the strongest association with smoking behavior? Adding odds ratios or regression coefficients would make this table more interesting.
―    Discussions: The emphasis on declining trends is strong, but potential biases in the underlying data, such as self-reported smoking rates, are not adequately emphasized (Lines 362–364). Emphasize potential data biases (e.g., underreporting) to balance the interpretation.
―    The discussions includ historical studies and does not integrate enough recent literature (within the last five years) to explore how contemporary smoking policies or health campaigns may influence the observed trends (Lines 396–403). Incorporate more recent studies or reports to frame the findings within the current public health landscape.
―    Potential inconsistencies in smoking definitions across the included studies are not compared against similar issues in international research. Highlight how variations in smoking definitions could affect cross-study comparisons and whether this issue aligns with international research challenges.
―    The discussion admits the variability in smoking definitions and the reliance on self-reported data as key limitations (Lines 360–362). The limitations of sampling biases in the included studies are briefly mentioned (Lines 157–164). Expand the limitations section to include: the potential impact of self-reported biases on smoking prevalence rates; the lack of statistical testing for trends and predictors in many reviewed studies; the uneven geographical and temporal representation of included studies.
―    The discussion highlights the novelty of focusing on Aboriginal health workers as a distinct subgroup and identifies them as a priority for future interventions (Lines 462–471). The paper does not adequately discuss how these insights advance the field or contribute to practical policy recommendations. The declining trends among healthcare professionals are positioned as encouraging, indicating the potential impact of broader public health policies (Lines 444–462). The discussion could better emphasize the unique contributions of the study, particularly in highlighting the need for tailored interventions. Explicitly state what this study adds to the field, particularly in relation to understanding predictors and subgroup disparities. Improve the discussion with practical implications for public health policies, including targeted interventions for high-risk groups like Aboriginal health workers.
―    Conclusions regarding predictors (e.g., workplace stress, cultural factors) are speculative and not directly supported by the evidence presented in the results (Lines 342–357). Clarify which conclusions are directly evidence-based and separate them from interpretations or hypotheses for future research.

Author Response

Thank you so much for reviewing our scoping review and providing a thorough and constructive feedback. We highly appreciate this. We have tried to address your comments to the best of our ability and we think this has significantly improved our revised manuscript. Please find below replies to specific comments made:

  1. The introduction establishes the premise that healthcare professionals who smoke are less likely to provide effective smoking cessation advice (Lines 1–13). This provides a logical basis for the study. It discusses the influence of smoking habits among healthcare professionals on public perception and health promotion roles, which aligns with the study's relevance to public health (Lines 15–20). The argument is diluted by general statements about healthcare professionals' responsibilities (e.g., lines 12–20). For example, the relationship between declining smoking rates and the remaining prevalence among specific healthcare groups is not articulated clearly enough. The authors do not sufficiently connect this data to previous studies in similar settings or regions. Including comparisons to international contexts and cite current data (within the past 5–7 years), such as updated latest global or national reports on smoking behaviours among healthcare workers would improve the introduction. The introduction does not clearly explain how the study differs from or improves past research. Clearly write a central argument summarizing the need to study smoking prevalence and predictors among healthcare professionals, emphasizing its relevance to Australia’s unique healthcare environment. Focus on the unique aspects of the argument (e.g., smoking prevalence trends and their impact on healthcare efficacy, recent public health campaigns, technological advancements in data collection, or shifting societal attitudes toward smoking in healthcare.

Reply:

We agree with your comments fully. Accordingly, the revised manuscript states a logical flow starting with the premise that healthcare professionals who smoke are less likely to provide effective smoking cessation advice, and then building up the case with statistics from international perspectives and recent trends in smoking among healthcare professionals and then bringing the uniqueness of the Australian healthcare system, where the knowledge gaps lie, and finally to how to address those gaps through the objectives of the review. Your valuable insights have been incorporated throughout the body of the introduction [Line: 28- 112].

  1. Lines 30–40: condense the repetitive details about the importance of healthcare professionals in smoking cessation.

Reply:

The repetitive details about healthcare about the importance of healthcare professionals in smoking cessation has been avoided now, condensed and revised accordingly in the new manuscript [Line: 30- 38].

  1. The paper emphasizes the innovative approach to studying smoking prevalence and predictors among Aboriginal Health Workers (AHWs), a subgroup often overlooked in research (Lines 47–60).  The study’s use of PRISMA and scoping review methodologies implies a novel methodological perspective.  But the originality of the analysed predictors is not underlined. So, emphasize the novel aspects of the study, such as its comprehensive scope, inclusion of underrepresented groups like AHWs (specific challenges they face regarding smoking cessation or the cultural context of smoking within Aboriginal communities), or innovative methodological approaches. Also connect the originality to actionable outcomes, such as informing tailored interventions or policy changes

Reply:

Thank you for this valuable comment. We have made an attempt to revise the introduction to address this issue, emphasizing the comprehensiveness of this review and the inclusion of the often overlooked high-risk AHWs group, and the actionable outcomes, both in the body of the introduction and in the very last paragraph [Line: 98-112]

  1. The claim that smoking prevalence data among Australian healthcare professionals is sparse (lines 41–50) is not convincingly supported by examples

Reply:

This is a good observation, indeed. This has been now addressed in the revised manuscript with examples [Line: 98-102]

  1. The aims, focusing on prevalence and predictors, are clearly stated at the end of the introduction (lines 61–64). However, their specificity feels unexpected, as the introduction does not clearly lead to these objectives; linking the discussion of gaps more directly to the aims would create a smoother transition.

Reply:

As mentioned earlier in reply 1, since we have made a significant change in the introduction section of the manuscript as per your suggestion, the new paragraphs clearly link with the aims and objectives of the review. We feel the transition to the objectives and research question from the discussion of gaps is much smoother now [Line: 28-111]   

  1. While the methodology is well-aligned with scoping reviews, the rationale for choosing this method over others (e.g., systematic review) is not explicitly stated.

Reply:

The rationale for choosing this scoping review method over others has now been addressed in the revised manuscript [Line 116-118]

  1. Can you provide a detailed justification for the selection of databases included in the literature search, particularly in relation to their relevance to the interdisciplinary nature of the research topic, which spans public health, psychology, and healthcare practices?

Reply:

Justification for the selection of multiple databases is now outlined in the revised manuscript as these capture the nursing, medical and psychological literature on smoking globally [Line: 184-187].

  1. The explanation of the screening and selection processes lacks sufficient detail. Although Covidence is recognized as a resource to simplify the procedure, providing more information on how inter-rater reliability was assessed (e.g., percentage agreement, kappa scores) would improve the study's transparency (Lines 125–133).

Reply:

Thanks for your valuable feedback. Just to reiterate that we have used endnote and covidence software to streamline the data review process and all three of the researchers were involved in every stages of screening. The percentage agreement for resolving disagreements has now been mentioned. So, we have made an attempt to improve the transparency now [Line:  201-210]  

  1. The confidence on self-reported data from the included studies introduces a risk of reporting bias. Nearly half of the studies did not mention a consistent definition of “current smoking status” (Lines 196–203), which may affect the reliability of aggregated findings.

Reply:

A valid observation, we did mention about this in the previous manuscript. However, this has been further strengthened now in the revised manuscript in the limitation section and the challenges it poses [Line: 719-734].

  1. Provide a more detailed discussion of potential limitations, particularly those related to selection bias, reporting bias, and geographic representation within the included studies.

Reply:

Thank you so much for this critical input. We have now addressed all these limitations in detail in the revised manuscript [719-734].

  1. While data extraction is mentioned, the process is not described in detail (Lines 124–133). For example, what specific variables were charted, and how were they categorized (e.g., demographic, psychological, environmental)? Clarify how data were extracted and whether a standardized coding scheme was applied to classify prevalence, predictors, or healthcare groups. If no statistical analysis was conducted, clearly state this to avoid confusion.

Reply:

            The data extraction process in the methodology section has now been modified based on this feedback. We mentioned how the different variables were used and why no statistical tests could be performed [Line: 201-210]

  1. Reporting of search terms and the Boolean operators used in each database search is not sufficiently detailed. Provide a complete list of search strategies used in each database, including exact terms, truncations, and Boolean combinations, either in the text or as supplementary material.

Reply:

This has been addressed and provided in supplementary material A.

  1. Figure 1 lacks detail on reasons for exclusion (e.g., how many papers were excluded due to non-relevance vs. insufficient data). Adding more specific categories would strengthen its utility.

Reply:

Figure 1 has been modified based on the feedback and now has a bit more details in regard to non-exclusion of the papers [Page: 10]

  1. Include annotations in Figure 2 (Smoking Trends) to highlight key patterns significant deviations, linking them to relevant societal, policy, or demographic factors.

Reply:

            Annotations and arrow marks have now been added to Figure 2, we think this figure is more understandable now. Thanks for your feedback [Page: 22].

  1. Table 2 (Prevalence of Smoking) and Table 4 (Study Characteristics) are too detailed and difficult to navigate. For instance, Table 2 includes excessive descriptive text that could be simplified.

Reply:

Thank you for your extremely valuable feedback. We have extensively worked on Table 2 and table 4. The first priority was to reduce the text and also to improve visual representation of the tables for easy navigation. We have omitted some columns like ‘Study design’ from Table 2 as most of the information is already described in the result section. In the same table we simplified the sample size in some of the studies and used a range to express the smoking prevalence. Also the smoking prevalence column has been brought forward in column 3 so it better describes the table. In table 4, we have removed the ‘data source and study period’ column, as again, this is expressed in the result section already. Current smoking definition column was modified also, so the entire table has now been condensed into 2 pages instead of the previous 4 pages [P 12: & P18]. 

  1. In Table 2, the smoking definitions vary across studies (e.g., daily smokers, occasional smokers), but these inconsistencies are not visually highlighted. This reduces the comparability of the presented data. Clearly define smoking statuses across all studies and consider using symbols or color codes to visually highlight variations.

Reply:

As mentioned in the immediate earlier response, definitions of smoking column have been modified now. We have only considered the current smoking and excluded the definitions for ever smoker or ex-smoker which was there before. Along with the omission of ‘data source and study period’ column’, we think the table is a bit more comparable now [P: 12].

  1. Table 3 (Predictors of Smoking) lists predictors but does not visually rank or quantify their significance, making it harder to assess which factors are significant. No statistical weight or ranking for the predictors is provided. For example, which predictor had the strongest association with smoking behavior? Adding odds ratios or regression coefficients would make this table more interesting.

Reply:

Based on the feedback, the entire table has now been changed. Previously it did not quantity the significance of the different predictors which were statistically significant. Different sets of predictor variables have been reorganized. The odds ratios and 95% confidence interval and p-values are now organized in a proper manner. This has significantly enhanced the visual outlook of the table. Ideally this could have been improved further if we had the OR and the 95% CI values for all the predictors from all the included studies which either showed a positive or a negative association with current smoking. Unfortunately, we could not construct a forest plot diagram to visually rank these predictors as we only had a few studies which narrated the OR and 95% CIs. The revised table, however, gives a much better perspective though [P: 15]  

  1. Discussions: The emphasis on declining trends is strong, but potential biases in the underlying data, such as self-reported smoking rates, are not adequately emphasized (Lines 362–364). Emphasize potential data biases (e.g., underreporting) to balance the interpretation.

Reply:

            Thanks for this feedback. We have worked on this and the revised manuscript talks about the potential data biases adequately in the limitation section [Line: 675-690].

  1. The discussions include historical studies and do not integrate enough recent literature (within the last five years) to explore how contemporary smoking policies or health campaigns may influence the observed trends (Lines 396–403). Incorporate more recent studies or reports to frame the findings within the current public health landscape.

Reply:

Thank you so much for your important feedback. We have mostly acknowledged your comments and have worked on these points to update the revised manuscript. However, please note the greatest challenge the authors faced to compare and contrast with contemporary data was not feasible because of paucity of data, not only from Australian context in the last 6-7 years but from the global stage as well. Even the few systematic reviews and meta-analyses that were carried out in recent times, data collection period for the included studies, be it for physicians or nurses or dentists or other allied health workers, were only up to 2012-2013 maximum. This has significantly reduced the comparable populations and settings discussion. We have tried to touch upon the disparities of the study findings also looked for possible causes for decline or a lack of decline in Australia and elsewhere. [Line: 483 - 778].

  1. Potential inconsistencies in smoking definitions across the included studies are not compared against similar issues in international research. Highlight how variations in smoking definitions could affect cross-study comparisons and whether this issue aligns with international research challenges.

Reply:

            Thanks for identifying this. In the revised manuscript, we have now tried to address this inconsistency by comparing it to a few other research findings in the limitation of the review section [Line: 736-749].

  1. The discussion admits the variability in smoking definitions and the reliance on self-reported data as key limitations (Lines 360–362). The limitations of sampling biases in the included studies are briefly mentioned (Lines 157–164). Expand the limitations section to include: the potential impact of self-reported biases on smoking prevalence rates; the lack of statistical testing for trends and predictors in many reviewed studies; the uneven geographical and temporal representation of included studies.

Reply:

Thanks again for highlighting this. As mentioned in the earlier responses, we have now expanded the limitation section to include all these biases as you have outlined [P: 719-734].

  1. The discussion highlights the novelty of focusing on Aboriginal health workers as a distinct subgroup and identifies them as a priority for future interventions (Lines 462–471). The paper does not adequately discuss how these insights advance the field or contribute to practical policy recommendations. The declining trends among healthcare professionals are positioned as encouraging, indicating the potential impact of broader public health policies (Lines 444–462). The discussion could better emphasize the unique contributions of the study, particularly in highlighting the need for tailored interventions. Explicitly state what this study adds to the field, particularly in relation to understanding predictors and subgroup disparities. Improve the discussion with practical implications for public health policies, including targeted interventions for high-risk groups like Aboriginal health workers.

Reply:

Thanks for your valuable feedback. We have made an attempt to address these in the revised manuscript, and have principally reflected them in the conclusion section [P: 759 – 778].

  1. Conclusions regarding predictors (e.g., workplace stress, cultural factors) are speculative and not directly supported by the evidence presented in the results (Lines 342–357). Clarify which conclusions are directly evidence-based and separate them from interpretations or hypotheses for future research

Reply:

Again, thank you for your valuable feedback. The conclusion section has been revised almost entirely to reflect an evidence based conclusion [P: 759-778].

Round 2

Reviewer 1 Report

Comments and Suggestions for Authors

The authors comprehensively addressed my comments and suggestions. So I have no further concerns. 

Author Response

REPLY TO REVIEW REPORT: REVIEWER 1

The authors comprehensively addressed my comments and suggestions. So I have no further concerns

Reply:

We would like to thank you once again for going through our 2nd version of the manuscript and for being satisfied with our revised manuscript. We highly appreciate your comment and feedback.

Wishing you a Happy New Year!

Reviewer 2 Report

Comments and Suggestions for Authors

―    Introduction: The revised version emphasizes the importance of focusing on Australian health professionals and highlights unique aspects, such as the inclusion of Aboriginal Health Workers (AHWs). Some connections to international contexts have been made. The introduction now transitions more clearly to the aims of the study, although the novelty of the predictors analyzed is not yet sufficiently emphasized.
―    Methods: The revised version elaborates on the selection of databases, highlighting their relevance to the interdisciplinary topic. Details on screening processes and inter-rater reliability (measured by percentage agreement) have been added. Though, the lack of statistical analysis is only mentioned, without further explanation. Variables have been grouped into general categories (demographic, psychological, and environmental), but additional details on coding schemes are missing.
―    Results: Tables have been reorganized but remain dense and insufficiently improved for clarity or comparability. While Table 2 is clearer, it does not visually highlight differences in smoking definitions, reducing its utility for comparison. Table 3 provides odds ratios (ORs) for some predictors, but it does not indicate which predictors are the most significant across studies. The discussion should highlight which predictors consistently exhibit the strongest ORs and their implications for targeted interventions. For instance, if male gender shows a high OR in multiple studies, its relevance should be emphasized in designing smoking cessation programs for healthcare professionals.
―    Discussion: The discussion now acknowledges biases in the data, such as self-reporting and inconsistent definitions, but does not explore their implications in depth. For example, does underreporting lead to an underestimation of smoking prevalence among healthcare workers? Do varying definitions across studies compromise the accuracy of trends or comparisons? The discussion highlights the novelty of focusing on AHWs but could be more analytical in connecting findings to practical policy recommendations. Public health implications should be addressed more concretely, particularly regarding how smoking among healthcare workers affects patient care and public perceptions.
―    Conclusions: The conclusions have been slightly rephrased for clarity. Although practical implications are discussed, they lack specificity and are not sufficiently tailored to predictors or subgroups (e.g., younger nurses, AHWs), which limits their practical applicability.

Author Response

REPLY TO REVIEW REPORT: REVIEWER 2

We would like to thank you once again for going through our 2nd version of the manuscript and for providing further valuable comments and inputs. We have found your comments to be unique and just, and felt like addressing these will surely improve the quality of our manuscript further and would make it more readable. We highly appreciate this. Like the previous round, we have made our best attempts to address all your comments. Please find below our line by line replies:

  1. Introduction

The revised version emphasizes the importance of focusing on Australian health professionals and highlights unique aspects, such as the inclusion of Aboriginal Health Workers (AHWs). Some connections to international contexts have been made. The introduction now transitions more clearly to the aims of the study, although the novelty of the predictors analyzed is not yet sufficiently emphasized.

Response:

Thanks for picking up this issue; the previous version indeed did not emphasize the predictors or elaborated the rationale of the 2nd objective linked with it in the introduction section. This latest revision talks about the novelty of the predictors in one full paragraph [Line: 54-71].  

  1. Methods:

The revised version elaborates on the selection of databases, highlighting their relevance to the interdisciplinary topic. Details on screening processes and inter-rater reliability (measured by percentage agreement) have been added. Though, the lack of statistical analysis is only mentioned, without further explanation. Variables have been grouped into general categories (demographic, psychological, and environmental), but additional details on coding schemes are missing.

Response:

Thanks for highlighting our efforts in streamlining this section in the last revision. In this latest revision, we have now addressed the reason for exclusion of statistical analyses [Line: 244-246]. We did also mention about the codes this time [Line: 240-242].

Moreover, in this revised manuscript, we have made attempts to improve the methodology section further, these included:

  1. How the overall search yield helped in understanding the topic [Line:220-223]
  2. An additional line on data screening [Line: 225-227]
  3. Additional line in regard to inter-rater reliability [Line: 233]
  4. Additional line on categories [Line: 236-237]

  5. Results:

Tables have been reorganized but remain dense and insufficiently improved for clarity or comparability. While Table 2 is clearer, it does not visually highlight differences in smoking definitions, reducing its utility for comparison. Table 3 provides odds ratios (ORs) for some predictors, but it does not indicate which predictors are the most significant across studies. The discussion should highlight which predictors consistently exhibit the strongest ORs and their implications for targeted interventions. For instance, if male gender shows a high OR in multiple studies, its relevance should be emphasized in designing smoking cessation programs for healthcare professionals.

Response:

We agree with your comment in this section. In this revised manuscript, we have tried to improve the tables for clarity/comparability.

  1. Table 2 did not deal with smoking definitions in the previous versions; it was always mentioned in Table 4. However, now we have now significantly modified Table 2 by adding a column on smoking prevalence by definitions. This column was sub-divided into further 4 columns clustering the different current smoking definitions and putting the values across these 4 columns respectively. This now gives better clarity in regard to current smoking prevalence [Page: 12-13].
  2. In Table 3, we have now standardized the p-values indicating power of associations for the studies that were available to either p<0.01, p<0.001 or p<0.05. [Page: 14]. This should help in better navigating this table. Only a few of the predictors were significant in more than one studies, and these are now additionally mentioned in the beginning of result section (3.2) that dealt with predictors of smoking [Line: 447-451]. The table itself with respective ORs or p-values is quite self-explanatory in regard to which association is stronger. However, as per your suggestion, we have now addressed this in the conclusion section in regard to their application in smoking cessation policies/interventions [Line: 824-832].    
  1. Discussion:

The discussion now acknowledges biases in the data, such as self-reporting and inconsistent definitions, but does not explore their implications in depth. For example, does underreporting lead to an underestimation of smoking prevalence among healthcare workers? Do varying definitions across studies compromise the accuracy of trends or comparisons? The discussion highlights the novelty of focusing on AHWs but could be more analytical in connecting findings to practical policy recommendations. Public health implications should be addressed more concretely, particularly regarding how smoking among healthcare workers affects patient care and public perceptions.

Response:

Thanks for your acknowledgement for our revised work as we have tried to address your concerns. Now we have further addressed your latest concerns.

  1. We have elaborated the implications for both self-reporting [Line: 793-798] and inconsistent definitions in the discussion section under limitations [Line: 775-777, 793-798].
  2. Connecting findings about AHWs to practical policy recommendations also addressed now [Line: 837-840].
  3. How smoking among healthcare workers affects patient care and public perceptions further addressed now in conclusion [Line: 817-821].
  1. Conclusions:

The conclusions have been slightly rephrased for clarity. Although practical implications are discussed, they lack specificity and are not sufficiently tailored to predictors or subgroups (e.g., younger nurses, AHWs), which limits their practical applicability.

Response:

Thanks for your comments. We have now further addressed these concerns in the revised conclusion and future directions sections respectively [Line: 808 – 874].
